# Primary Human Trabecular Meshwork Model for Pseudoexfoliation

**DOI:** 10.3390/cells10123448

**Published:** 2021-12-07

**Authors:** Munmun Chakraborty, Prity Sahay, Aparna Rao

**Affiliations:** 1Hyderabad Eye Research Foundation (HERF), L.V. Prasad Eye Institute, Bhubaneswar 751024, India; munmun282@gmail.com (M.C.); pritpyari@gmail.com (P.S.); 2School of Biotechnology, KIIT University, Bhubaneswar 751024, India

**Keywords:** pseudoexfoliation, in vitro cell line model, human trabecular meshwork, TGF-β1, fibrosis

## Abstract

The lack of an animal model or an in vitro model limits experimental options for studying temporal molecular events in pseudoexfoliation syndrome (PXF), an age related fibrillopathy causing trabecular meshwork damage and glaucoma. Our goal was to create a workable in vitro model of PXF using primary human TM (HTM) cell lines simulating human disease. Primary HTM cells harvested from healthy donors (*n* = 3), were exposed to various concentrations (5 ng/mL, 10 ng/mL, 15 ng/mL) of transforming growth factor-beta1 (TGF-β1) for different time points. Morphological change of epithelial–mesenchymal transition (EMT) was analyzed by direct microscopic visualization and immunoblotting for EMT markers. Expression of pro-fibrotic markers were analyzed by quantitative RT-PCR and immunoblotting. Cell viability and death in treated cells was analyzed using FACS and MTT assay. Protein complex and amyloid aggregate formation was analyzed by Immunofluorescence of oligomer11 and amyloid beta fibrils. Effect of these changes with pharmacological inhibitors of canonical and non-canonical TGF pathway was done to analyze the pathway involved. The expression of pro-fibrotic markers was markedly upregulated at 10 ng/mL of TGF-β1 exposure at 48–72 h of exposure with associated EMT changes at the same time point. Protein aggregates were seen maximally at these time points that were found to be localized around the nucleus and in the extracellular matrix (ECM). EMT and pro-fibrotic expression was differentially regulated by different canonical and non-canonical pathways suggesting complex regulatory mechanisms. This in vitro model using HTM cells simulated the main characteristics of human disease in PXF like pro-fibrotic gene expression, EMT, and aggregate formation.

## 1. Introduction

Pseudoexfoliation is an age related fibrillopathy characterized by abnormal fibrillar extracellular material (ECM) in ocular tissues [1,2]. Pseudoexfoliative aggregate material causing mechanical obstruction of the trabecular meshwork (TM), blood aqueous barrier dysfunction [1,2], endothelial cell dysfunction, and abnormal ECM homeostasis cause TM dysfunction/fibrosis eventually leading to glaucoma, if untreated. The pseudoexfoliative material comprises of non-collagenous basement membrane components such as laminin, fibronectin, amyloid P, and vitronectin as well as proteinaceous components of elastic fibres (such as elastin, tropoelastin, fibrillin-1, microfibril-associated glycoprotein-1) and latent TGF-β-binding proteins (LTBP-1 and -2) [3]. Transforming growth factor, tissue matrix metalloproteinases (MMPs) [1,2,3] and plasminogen activator inhibitor-1 (PAI-1) regulate ECM homeostasis with increased PAI-1 levels causing excessive ECM deposition and reduced degradation in adjoining tissues [4]. While Lysyl oxidase homolog 1 (LOXL1) is deemed necessary for disease pathogenesis [1,5], it is now understood that LOXL1 alone does not explain the preferential geographical distribution or the differential role of different genes in disease pathogenesis or glaucoma onset in different ethnic populations. It is well recognized that environmental factors, epigenetics, and their interplay with gene expression is what may hold the key for explaining the disease pathogenesis.

These are the challenges to successful consistent replication of the disease in an animal or an in vitro model for pseudoexfoliation. Currently, mice harboring a spontaneous mutation in the lysosomal trafficking regulator (*LYST*) gene have been shown to have closest feature of human disease in the form of iris transillumination defects [5]. Yet, glaucoma was not seen in these mice despite presence of fibrillin positive aggregates in the anterior chamber. Another group reported ocular features consistent with PXF like disorganized ciliary body and accumulation of exfoliation like extracellular material in the anterior segment in mice injected with recombinant adenovirus coding Wnt5a. Yet, presence of glaucoma could not be confirmed in this model [6].

Several groups have shown that the aqueous concentrations of TGF-β1, were higher in eyes with pseudoexfoliation syndrome (PXF) and pseudoexfoliation glaucoma (PXG) than controls or other primary glaucoma [7,8]. Increased levels of few cytokines, namely, interleukin 6 (IL-6), interleukin 8 (IL-8), tumor necrosis factor-alpha (TNF-α), and vascular endothelial growth factor (VEGF) have also been reported in the aqueous humor of glaucomatous eyes [9,10]. These pro-inflammatory cytokines have been shown to favor and create a pro-fibrotic tissue milieu [9,10,11,12] and with oxidative stress they are known to play a role in PXF and PXG [12,13]. In our previous study, we found elevated TGF-β1 levels in the aqueous humor of patients that increased with disease stages, with maximal levels being found in eyes with ocular hypertension and glaucoma [14]. We also found decreasing tear and aqueous MMP9 activity, with increased inflammatory cytokine levels, in severe PXG eyes compared to primary glaucoma [14], suggesting that the elevated TGF levels somehow were important for molecular events that causes the disease. We also found serum levels to parallel the levels in the aqueous and tears with several key miRNA driven by TGF-β1 to be upregulated in eyes with ocular hypertension (OHT) and PXG [15]. This was also associated with downregulation of several unfolded protein response (UPR) genes, a key mechanism of abnormal ECM homeostasis in PXF. This study now explores the molecular effects of sustained TGF-β1 exposure of HTM cell in vitro and to evaluate expression of ECM/elastic fiber components as well as morphological/molecular changes induced thereby.

## 2. Materials and Methods

### 2.1. Isolation and Culture of TM Cells

Primary Human Trabecular meshwork (HTM) cells were established from healthy cadaver eyes (*n* = 3) obtained from the institutional eye bank within 6 h of death using standard methods described elsewhere [16]. Briefly, the trabecular meshwork (TM) stripped from the procured corneoscleral rims (see Appendix A) were thoroughly washed with Dulbecco’s-phosphate buffer saline (D-PBS, A12856-01, Gibco, Carlsbad, CA, USA) followed by dissection of the TM tissue into tiny pieces with microscissors under a dissection microscope (Olympus SZX7, Tokyo, Japan). The cut TM pieces were digested with collagenase treatment (1 mg/mL at 37 °C for 2 h) which was followed by centrifugation (1200 rpm, 5 min) and trypsinization (0.25% trypsin, 5 min). The TM cells were pelleted down by centrifugation (1200 rpm, 5 min) and plated on a 35 mm dish with endothelial cell basal medium, EBM-2 media (cc-3156, Lonza, Basel, Switzerland) at 37 °C and 5% CO_2_. Only three-fifth passage cells of 50–70% confluence were used for all study experiments after validation of each cell lineage by upregulation of myocilin expression after dexamethasone treatment [17].

### 2.2. TGF Treatment and Cell Viability Assays

Cultures that reached 50–70% confluence in flasks were serum starved for 24 h, followed by treatment with TGF-β1 (T7039, Sigma Aldrich, St. Louis, MO, USA) at different concentrations (3 ng/mL, 5 ng/mL, 10 ng/mL, 15 ng/mL) for 24 h, 36 h, 48 h, 72 h, and 96 h while one set, that was left untreated, was used as control. Cell viability in control and treated cells was measured using the 3-(4,5-dimethylthiazol-2-yl)-2,5-diphenyltetrazolium bromide (MTT, Sigma-Aldrich, St. Louis, MO, USA) assay using a previously described procedure [14]. Cells were serum starved for 24 h and then treated with or without TGF-β1 (3 ng/mL, 5 ng/mL, 10 ng/mL, 15 ng/mL) for 24 h, 36 h, 48 h, 72 h, and 96 h. Absorbance at 570 nm was measured on an EPOCH microplate reader (BioTek, Winooski, VT, USA).

To evaluate the mechanism of recruitment of profibrotic markers by TGF-β1 treatment (canonical and non-canonical pathways), cells were pre-treated for 1 h with small molecule inhibitors, 2.5 µM SMAD3 inhibitor (SIS3, Santacruz Biotechnology, Dallas, TX, USA), 10 µM ERK inhibitor (328006, Calbiochem, Sigma Aldrich, St. Louis, MO, USA), 10 µM JNK inhibitor II (420119, Calbiochem, Sigma Aldrich, St. Louis, MO, USA) and 10 µM p38 MAP Kinase inhibitor (506172, Calbiochem, Sigma Aldrich, St. Louis, MO, USA) before TGF-β1 treatment in a separate set of experiments. Lysates of TM cells treated with TGF-β1, with or without pretreatment with inhibitors, were used for different experiments.

Apoptotic cell death was analyzed in treated and untreated cultures using the annexin V-fluorescein isothiocyanate (FITC)/prodium iodide (PI) dual staining method with the Annexin V-FITC Apoptosis Detection kit (MintenyiBiotec, Bergisch, Gladbach, Germany) and were analyzed using the BD FACS Canto II Flow Cytometer (BD Biosciences, San Jose, CA, USA).

### 2.3. Epithelial Mesenchymal Transition (EMT) and Morphological Changes

Serum starved and TGF-β1 treated (0 ng/mL, 5 ng/mL, 10 ng/mL, and 15 ng/mL) HTM cells were analyzed for morphological changes in shape and structure at 24 h, 48 h, and 72 h after treatment. EMT was indicated when HTM cells became elongated, spindle-like, and densely packed losing typical characteristic morphology of HTM cells. These changes were captured using an inverted microscope (Olympus CKX53, Tokyo, Japan), which was confirmed with upregulation of smooth muscle actin, α-SMA and vimentin, which are markers for mesenchymal cells (α-SMA is more specific for myofibroblast cells. During EMT, cells (including HTM cells) acquire myofibroblast phenotype leading to expression of α-SMA). Percent of total cells undergoing morphology change was also quantified using imageJ software. Since the cells become spindle shaped after treatment, there were two criteria of measurement, change in length of cells and length between cells (since cells become densely packed the length between cells decreases). To quantify, an area of 310 cm^2^ was selected on the micrographs and the average length of control cells was calculated and was set as standard. The length of treated cells was compared to the standard length and the number of cells whose length exceeded the standard value was calculated and plotted as percent of total cells. Similarly, we calculated the length between cells in control and treated cells and the average length was plotted.

### 2.4. Gelatin Zymography Analysis for MMP9

Enzymatic activity of MMP-9 was examined by substrate gelatin zymography as described previously [14]. Protein quantification was done by Bradford assay and equal quantity of supernatant media protein from control and treated HTM cells were separated on 10% SDS-PAGE gels containing 0.1% gelatin. The lysis zones were analyzed using ImageJ and the MMP9 activity was expressed in arbitrary units (AU).

### 2.5. RNA Isolation and Quantitative Polymerase Chain Reaction (qPCR)

Total RNA from cultured cells was extracted using QIAzol lysis reagent (QIAGEN, Hilden, Germany). One microgram of RNA was reverse-transcribed using Reverse Transcriptase core kit (RT-RTCK-03, EUROGENTEC, Liege, Belgium) and the cDNA was subjected to quantitative PCR (for Primers whose sequences are listed in Table 1) using PowerUp SYBR Green qPCR Master Mix (A25741, Appliedbiosystems, Foster city, CA, USA). Ct values were normalized to *GAPDH*, which served as an internal control and results were analyzed using the ΔΔ threshold (Ct) method.

### 2.6. Immunoblotting

Immunoblotting was done by a method as described previously [14]. Protein quantification was done by Bradford assay and equal quantity of protein was loaded onto each lane. The primary antibodies used in the study were mouse anti-α-SMA (ab7817, 1:1000, Abcam, Cambridge, MA, USA), anti-vimentin (D21H3, 1:1000, Cell Signalling Technology, Beverly, MA, USA) rabbit anti-fibronectin (AF5335, 1:1000, Affinity Biosciences, Brisbane, Queensland, Australia), mouse anti-fibulin-5 (ab66339, 1:800, Abcam, Cambridge, MA, USA), mouse anti-PAI-1 (ab125687, 1:1000, Abcam, Cambridge, MA, USA), rabbit anti-Fibrillin (AF0429, 1:1000, Affinity Biosciences, Brisbane, Queensland, Australia) and rabbit anti-pSMAD3(C25A9, 1:1000, Cell Signalling Technology, Beverly, MA, USA). Rabbit anti-GAPDH (1:10,000; Abcam, Cambridge, MA, USA) was used as the loading control.

### 2.7. Immunofluorescence Assay for Fibrillar Aggregates

Control and treated cells were seeded in coverslips coated with 0.1% gelatin before TGF-β1 treatment for this experiment. The cell culture medium was separated, and the cells were washed in PBS followed by fixation with 4% paraformaldehyde for 15 min at room temperature (RT). Permeabilization of the cells was done with 0.2% Triton X-100 solution for 10 min followed by incubation with blocking solution (5% FBS in 1X PBS) for 20 min at RT with gentle rocking to block non-specific sites. Cells were then incubated with primary antibodies, Oligomer 11 (AHB0052, 1:1500, Invitrogen, Beverly, MA, USA) and amyloid fibril (PA5-77843, 1:1500, Invitrogen, Beverly, MA, USA) overnight at 4 °C, followed by incubation with secondary antibody (anti-rabbit alexa 488 1:2000, Invitrogen, Beverly, MA, USA) mixed with 2 µg/mL Hoechst (for nucleus staining) for 1 h at RT. The coverslips were mounted and observed under fluorescent microscope (Olympus, BX53, Tokyo, Japan).

### 2.8. Statistics

Each experiment was repeated in triplicates. The mean data were expressed as mean +/− SD. Statistical analysis was performed using Graphpad prism (version 7, San Diego, CA, USA). The data was drawn from normally distributed populations so parametric tests were used. Unpaired student *t*-test was used to determine statistical difference between two groups and One-way ANOVA was used for >2 groups with post hoc analysis with <0.05 considered statistically significant as indicated in figures.

## 3. Results

### 3.1. Continued Exposure of 10 ng/mL of TGF—β1 Causes Time Dependent Loss of Cell Viability and Cell Death in HTM Cells

The TGF-β1 signaling pathway is an extremely versatile and complex pathway that regulates cell survival and repair in different cell types with different functions and in different contextual situations. Context-driven specificity of this pathway is what determines the final fate of cells. Exploring its role on HTM survival, we treated HTM cells with different concentrations of TGF-β1 for different time-points and evaluated cell viability and cell death as described previously. In MTT assay, the cell viability of treated cells was found to decrease in a time dependent manner at all concentrations > 5 ng/mL, with a prominent decrease seen at 36–48 h, Figure 1. Concentration of 3 ng/mL seemed to cause minimal changes on HTM cells viability even up to 72 h (Figure 1a). Since not much of a change was observed in the cell viability from 36 h to 72 h with most consistent effect was seen with 10 ng/mL, we restricted our further experiments to 24 h, 48 h, and 72 h time points only. This correlated with increased in relative apoptotic cells commencing at 24 h on FACS compared to untreated cells (since control cells will also have some amount of apoptosis, the total percent of apoptotic cells (early and late apoptotic) was plotted as relative apoptosis (% of control)) with a peak seen at 72 h with 10 ng/mL concentration (Figure 1b). This effect was less prominent with 5 ng/mL TGF-β1 concentration (see Appendix A).

### 3.2. TGF-β1 Induces Time Dependent EMT in HTM Cells

After treatment, the HTM cells were observed to change shape to a confluent, epithelial-like monolayer of closely-attached elongated, flattened, spindle shaped cells suggestive of EMT (see EMT and morphology changes in methodology section). This was seen to commence at 48 h and continued till 72 h where the cells were more densely packed and appeared more flattened (Figure 2a). The morphological changes were quantified using ImageJ based on quantifiable morphology change. The percent of total cells undergoing morphological change began at 48 h and was found to be more pronounced with 10 ng/mL concentration while lower concentrations required longer exposure time for same morphological effect (see Appendix A).

### 3.3. TGF-β1 Causes a Time-Dependent Reduction in MMP9 Enzyme Activity

MMP9 remains an important modulator of TGF mediated regulation of ECM homeostasis and degradation. While the pro form of MMP9 increased at 72 h, the MMP9 enzyme activity was reduced progressively with sustained TGF exposure, (Figure 2b). This, however, was in contrast with increased MMP9 mRNA expression in treated cells at 72 h at all concentrations, (Figure 2d). These results suggested that though there is a concentration and time dependent upregulation of MMP9 gene expression, and the pro-forms induced by TGF, the enzyme activity was directly inhibited in a time-dependent manner on continued exposure.

### 3.4. TGF-β1 Induced Expression of Pro-Fibrotic and ER Stress Markers

Since the pseudoexfoliative aggregate are mainly composed of fibrotic and elastic components, we investigated the expression of *α-SMA*, *COL6A2*, *FBN1*, *FN1*, and *PAI-1* in TGF-β1 treated cells. We found that the mRNA expression and protein levels of all the above pro-fibrotic molecules increased after 10 ng/mL of TGF-β1 treatment compared to control at 48–72 h (Figure 3a–d). The same effect was seen at higher concentrations of 15 ng/mL (see Appendix A).

Lower concentration of 5 ng/mL showed asymmetric changes with different molecules getting upregulated at different time points suggesting the pro-fibrotic activity was maximally upregulated at 10 ng/mL of TGF-β1 after 24 h of sustained exposure. Since misfolded proteins seen in pseudoexfoliation recruits the Unfolded protein response (UPR) pathway and the inflammatory cytokines, we evaluated the expression of UPR genes X-box-binding protein 1 (*XBP1*) and Chaperon containing TCP1 subunit 4 (*CCT4*) that we found earlier to be downregulated in patients with advanced pseudoexfoliation glaucoma [15]. Consistent with our earlier study, we found *XBP1* and *CCT4* to be downregulated in vitro after 48 h exposure of 10 ng/mL TGF-β1 (Figure 3f). This was associated with raised expression of pro-inflammatory cytokines *IL-6* and *IL-8* for 24–72 h (see Appendix A).

### 3.5. Continued TGF-β1 Exposure Induces Fibrillar Aggregate Formation in HTM Cells In Vitro

The major characteristic of pseudoexfoliation syndrome is fibrillar aggregates that accumulate in the ECM. Therefore, we analyzed for oligomer11 and amyloid fibrils, key precursors of protein aggregate complex, in cells treated with TGF-β1. These were visualized maximally at 72 h with oligomer11 localized near the nucleus (Figure 4a) while amyloid fibrils were spread extensively throughout the cell (Figure 4b).

### 3.6. TGF-β1 Induces Downstream Changes on HTM Cells and ECM Using Both Smad and Non-Smad Signalling Pathways

To elucidate the signaling pathways involved for pro-fibrotic changes and EMT we further treated the cells with TGF-β1 (10 ng/mL) with and without various canonical and non-canonical TGF-β1 inhibitors. SIS3 which inhibits the Smad3 signalling pathway (phospho smad3 levels were ana-lysed to confirm involvement of Smad signalling pathway, see Appendix A) caused reduced TGF induced fibronectin expression, whereas expression of other ECM proteins did not change significantly with SMAD3 inhibition. While EMT changes (characterized by changes in α-SMA levels and morphological transformation) was prevented at 48 h following treatment of Smad3, ERK and p38 MAPK inhibitors, (Figure 5). JNK inhibition had no effect on α-SMA levels or other fibrotic markers. Inhibition of non-canonical pathways including ERK causes reduced expression of fibronectin and fibrillin with JNK inhibition being the sole pathway causing reduced Fibulin-5 expression. These results suggest that the expression of downstream pro-fibrotic markers involves a complex regulation by both canonical and non-canonical pathways regulating different proteins.

## 4. Discussion

Excessive accumulation of ECM proteins is considered a major feature of glaucomatous TM tissues. The TM cells and surrounding ECM work like a syncytium complex called matrisome that regulates aqueous humor outflow in the anterior chamber [18]. The pathogenic mechanism of increased resistance to aqueous humor outflow in PXF is poorly understood, but excessive ECM deposition, aggravated cell death, and alterations in cytoskeletal organization [1,2,8] apart from mechanical obstruction of the TM spaces by exfoliative material, have been thought to be the key mechanisms. Experimental models to study these mechanisms are lacking with current animal models mimicking certain ocular features of the human disease without evident glaucoma [3,19,20]. One of the key feature of PXF is protein aggregate formation. The exact composition of the pseudoexfoliative material is still not known histological and biochemical approaches have shown that the aggregates in PXF have a protein core surrounded by glycoconjugates that are resistant to enzymatic degradation [19]. The protein core has been shown to be composed of ECM proteins, MMPs and cross-linking enzymes like LOXL1 [3,4,18,19]. Therefore, we tried to create an in vitro model on HTM cells that will mimic PXF conditions in the eye. In this in vitro model, TGF-β1 (10 ng/mL) exposure caused upregulated expression of ECM proteins, EMT morphological changes, paralleled by deregulated enzymatic activity of MMP9 and protein aggregate formation at 48–72 h of exposure, all of which mimic features of PXF disease in the human eye. We visualized the protein aggregates using antibodies that bind to amyloid fibrils, key component of misfolded proteins. Though transmission electron microscope (TEM) and a 3D model of HTM cell culture would have been ideal, nanometer size of the aggregates and disruption during processing prevented visualization of these fibrils. Here we found that chronic TGF-β1 (10 ng/mL) exposure induces EMT in TM cells at 48 h, which was also confirmed by increased α-SMA and vimentin expression. This paralleled increased IL-6 and IL-8 levels at the same time period, both of which are TGF-induced deleterious effects for ECM homeostasis [12,21,22]. TGF-β1 induced EMT was inhibited by Smad3 inhibition, while other proteins were differentially regulated by other pathways. TGFβ1 is known to orchestrate EMT as a mechanism of regulating ECM homeostasis and cell survival [23,24,25,26]. EMT is a normal process involved in tissue repair [27,28] but deregulated EMT can lead to fibrosis. EMT-associated disease pathogenesis has already been studied in a range of diseases including lung fibrosis and cancer [29]. In the eye, EMT has been associated with subretinal fibrosis leading to age-related macular degeneration (AMD) [25]. The canonical and non-canonical pathways differentially regulate ECM proteins and this maybe one of the mechanisms of the TGF paradox driving increased fibrosis and functional damage in eyes with glaucoma in pseudoexfoliation. Study of these effects in our model particular could give more insights into actual regulation and cross talk of other pathways modulating TGF induced fibrosis in HTM cells. It may also be used to identify putative molecules to reverse EMT, cell death or aggregate formation, which may form important targets for preventing glaucoma in pseudoexfoliation.

To maintain homeostasis, cells rely on protective mechanisms to help them cope with ER stress, pathways referred to collectively as the unfolded protein response (UPR) [30]. The relation between UPR signaling and fibrogenesis are not fully understood. Studies suggest that ER stress facilitates fibrotic remodeling through activation of pro-apoptotic pathways, induction of epithelial—mesenchymal transition, and promotion of inflammatory responses [30,31,32]. In our previous study, we found that the expression of UPR response genes like *XBP1* and *CCT4* were significantly downregulated in PXG [15]. Therefore, we wanted to see if the UPR pathway is also hampered in the in vitro model. We found reduced expression of UPR markers like *XBP1* and *CCT4* at delayed TGF-β1 exposures. This hints at the insufficient UPR activation hampering the process of protein clearance and the misfolded proteins forming insoluble aggregates. Overall, our data indicate that sustained TGF-β1 treatment used in this in vitro HTM cell lime model could be used to mimic PXF human disease. This may enable studying temporal molecular events at different time points and allow us to identify drug targets for preventing cell death or aggregate formation.

## Figures and Tables

**Figure 1 cells-10-03448-f001:**
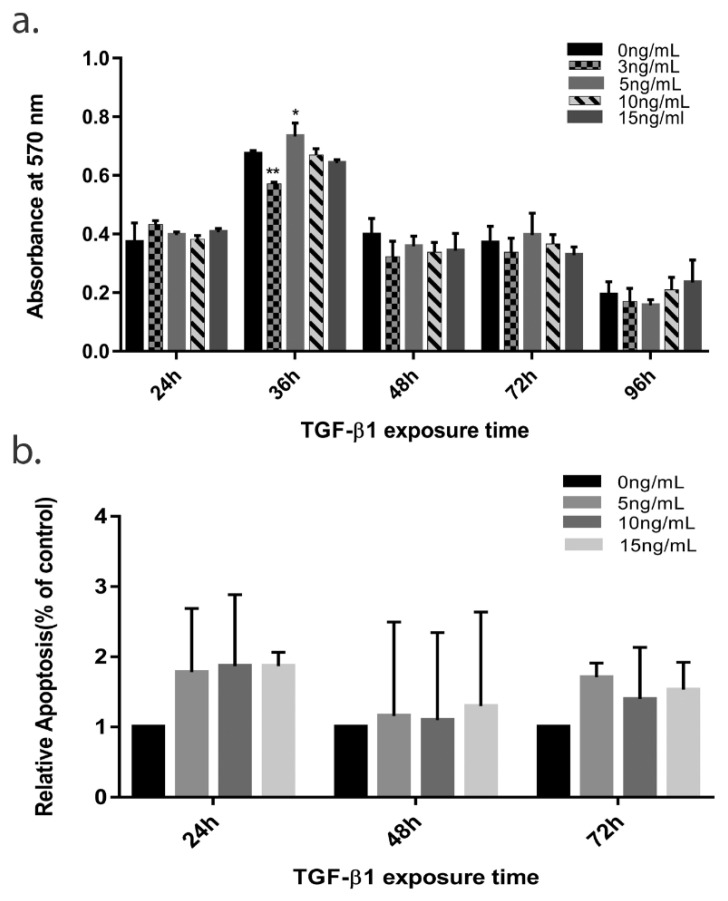
**The effect of TGF-β1 on TM cell viability and apoptosis.** (**a**) TM cell viability for 96 h with or without TGF-β1 treatment (3 ng/mL, 5 ng/mL, 10 ng/mL, 15 ng/mL). (**b**) Total apoptotic cells (early and late) plotted as Relative apoptosis (% control) against time at various TGF-β1 treatment. The error bars represent standard deviation (* *p* < 0.05, ** *p* < 0.01 by one way ANOVA).

**Figure 2 cells-10-03448-f002:**
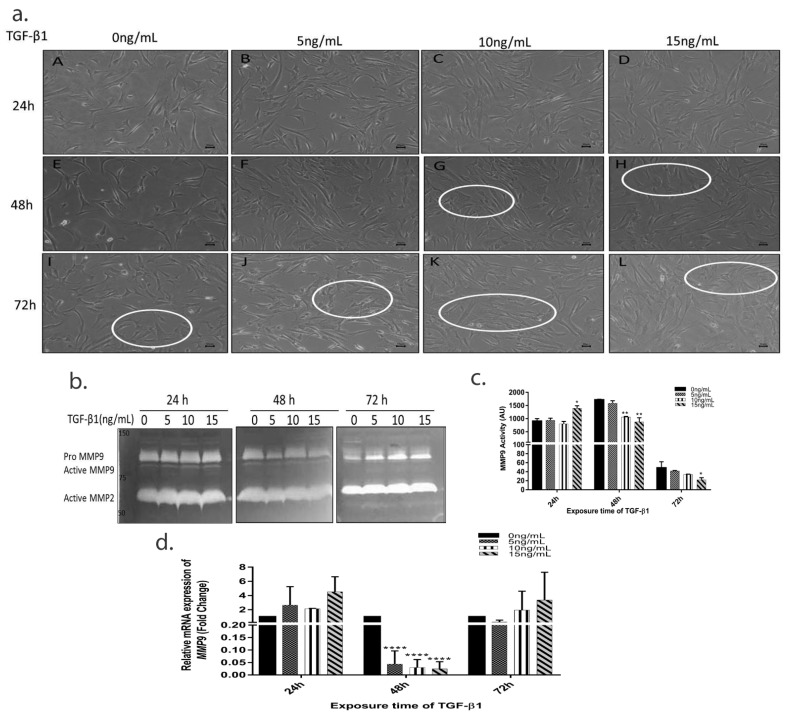
**TGF-β1 induces morphological changes and effects functional activity of MMP9.** (**a**) Panel (**A**)–(**L**) depicts representative photomicrographs of TM cells after TGF-β1 treatment. Circles show flattened, elongated, spindle shaped morphology of TM cells representing EMT. Bar = 50 µM. (**b**) MMP9 activity visualized by gelatin zymography (**c**) Quantification of MMP9 activity in arbitrary units using ImageJ, (**d**) relative mRNA expression of MMP9 gene after TGF-β1 treatment. The error bars represent standard deviation (* *p* < 0.05, ** *p* < 0.01, **** *p* < 0.0001, by one way ANOVA).

**Figure 3 cells-10-03448-f003:**
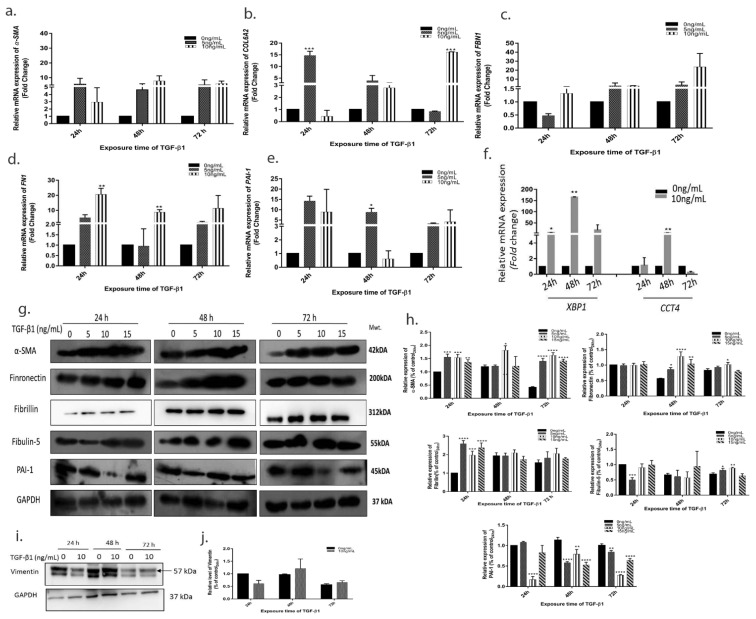
**TGF-β1 induces expression of pro-fibrotic molecules**. Relative mRNA expression of (**a**) *α-SMA* (**b**) *COL6A2* (**c**) *Fibrillin* (**d**) *Fibronectin* (**e**) *PAI-1* (**f**) UPR genes *XBP1* and *CCT4* (**g**) Protein expression of pro-fibrotic molecules after TGF-β1 treatment with densitometry analysis using ImageJ (**h**). (**i**) Vimentin expression after 10 ng/mL TGF-β1 treatment and densitometry analysis (**j**). The error bars represent standard deviation (* *p* < 0.05, ** *p* < 0.01, *** *p* < 0.001, **** *p* < 0.0001, by one way ANOVA).

**Figure 4 cells-10-03448-f004:**
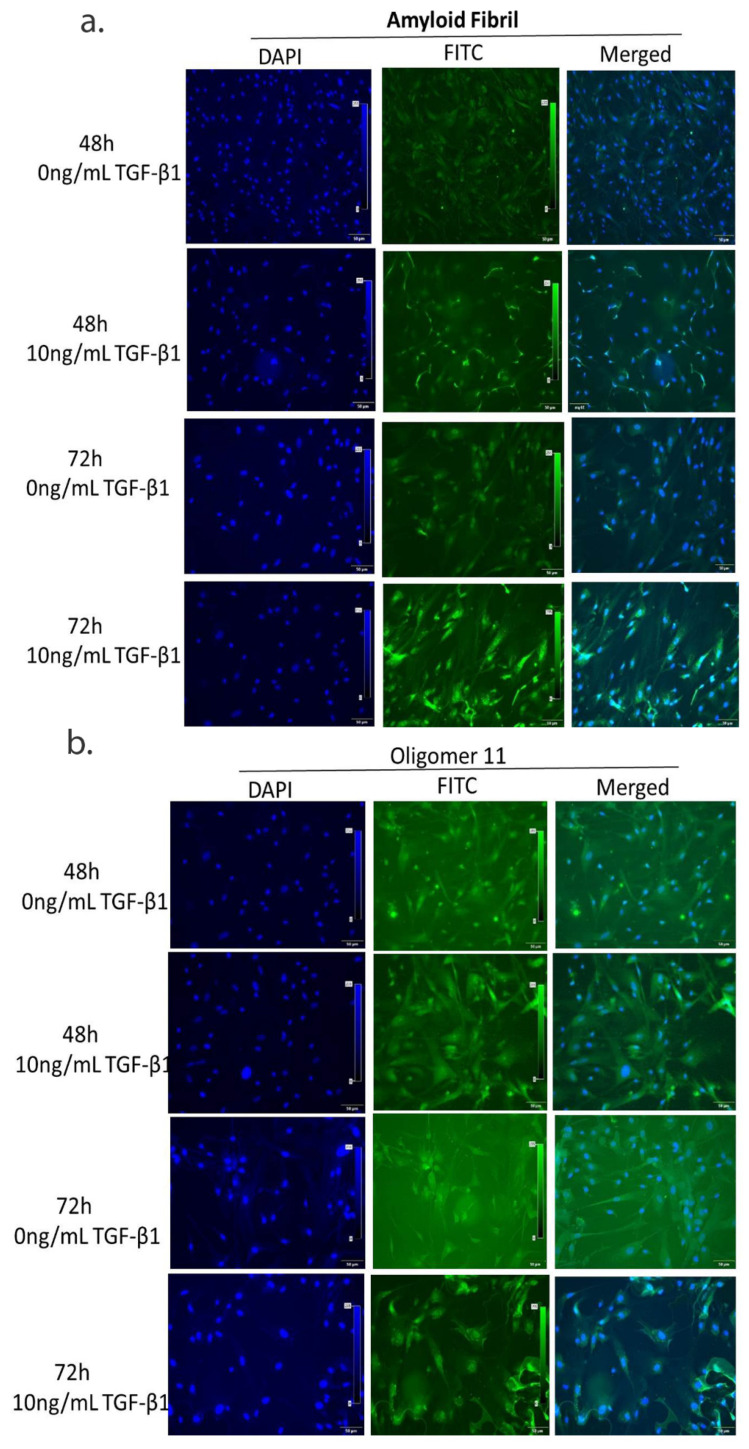
**TGF-β1 causes fibrillar aggregates in TM cells**. Immunofluorescence images of protein aggregates. Misfolded protein in cultured TM cells after TGF-β1 (10 ng/mL) treatment for 48 h and 72 h stained with (**a**) Amyloid fibril antibody, (**b**) oligomer 11 antibody (at 20× magnification). Bar = 50 µM.

**Figure 5 cells-10-03448-f005:**
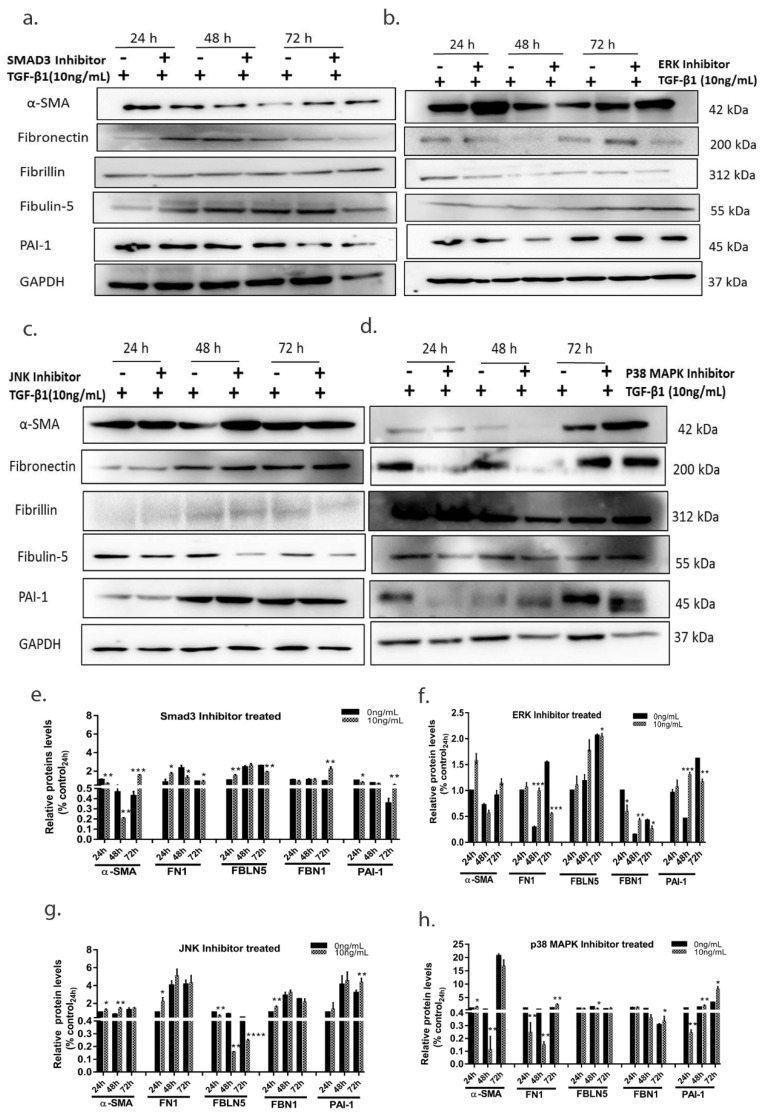
**The effect of pharmacological inhibition of TGF canonical and non-canonical pathways.** Cells treated with TGF-β1(10 ng/mL) and with or without inhibitors were analyzed for various fibrotic protein by Immunoblotting blot after (**a**) SMAD3 inhibition. (**b**) ERK inhibition (**c**) JNK inhibition (**d**) p38 MAPK inhibition. Differential effects of pharmacological inhibition was seen on pro-fibrotic protein expression, (**e**–**h**) Representative densitometry analysis of the protein bands. The error bars represent standard deviation (* *p* < 0.05, ** *p* < 0.01, *** *p* < 0.001, **** *p* < 0.0001, by *t*-test).

**Table 1 cells-10-03448-t001:** Quantitative PCR Primer sequences list.

Gene	Forward Sequence	Reverse Sequence
*α-SMA*	GAAGGAGATCACGGCCCTA	ACATCTGCTGGAAGGTGGAC
*COL6a2*	AGAGCTGTCCTTCGTGTTCCT	CTGTCATAGTCCTTCTCGTGGAA
*FBLN5*	CCTGTTCCGCTGTGAGTG	ACTGATGCACGTGGTTGG
*FN1*	CCACCCCCATAAGGCATAGG	GTAGGGGTCAAAGCACGAGTCATC
*PAI-1*	GGCCATTACTACGACATCCTG	GGTCATGTTGCCTTTCCAGT
*GAPDH*	GAAGGTGAAGGTCGGAGTC	GAAGATGGTGATGGGATTTC
*XBP1*	TCATGGCCTTGTAGTTGAGA	GGCATTTGAAGAACATGACTGG
*CCT4*	TGCTTTTGCAGATGCTATGG	GGACAACCAGTTCCTCCAAA
*IL-6*	CAAATTCGGTACATCCTCGACGGC	GGTTCAGGTTGTTTTCTGCCAGTGC
*IL-8*	CCACCGGAAGGAACCATCTCAC	GGCAAAACTGCACCTTCACACAG

## Data Availability

Not applicable.

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
