# Peer review of "Primary Human Trabecular Meshwork Model for Pseudoexfoliation"

_cells, 2021, doi:10.3390/cells10123448_

Round 1

Reviewer 1 Report

  1. Important! Expert statistical consultation needed. Is the data drawn from normally distributed populations” required for use of the student t-test and ANOVA tests? If not, then non-parametric tests should be used throughout this study. This must be verified.
  2. Abstract “found to be localised at around the nucleus” should be changed to “found to be localized at the nucleus” or “found to be localized around the nucleus”. Which is it?
  3. Intro line 43-44 “Transforming growth factor, tissue matrix metalloproteinases (MMPs)[1,2,3] and plasminogen activator inhibitor-1(PAI-1) regulates ECM homeostasis…” should be “Transforming growth factor, tissue matrix metalloproteinases (MMPs)[1,2,3] and plasminogen activator inhibitor-1(PAI-1) regulate ECM homeostasis…” regulate should be pleural.
  4. Line 47 to 50. Avoid using “It is not understood” twice within a few lines.
  5. Intro line 54-55. “Currently, mice harbouring a spontaneous mutation in the lysosomal trafficking regulator (LYST) gene has been shown…” should be “Currently, mice harboring a spontaneous mutation in the lysosomal trafficking regulator (LYST) gene have been shown….”
  6. Intro lines 58 to 60. Change to this. “Another group reported ocular features consistent with PXF like disorganized ciliary body and accumulation of exfoliation like extracellular material in the anterior segment in mice injected with recombinant adenovirus coding Wnt5a. Yet, the presence of glaucoma could not be confirmed in this model.”
  7. Intro lines 62 to 66. “Several groups (these two words are a different font size) have shown that the aqueous concentrations of TGF-β 1 were higher in eyes with pseudoexfoliation syndrome (PXF) and pseudoexfoliation glaucoma (PXG) than controls or other primary glaucoma [7,8]. Increased levels of a few cytokines, namely, interleukin 6 (IL-6), interleukin 8 (IL-8), tumor necrosis factor-alpha (TNF-α), vascular endothelial growth factor (VEGF) have also been reported…” Note the names of the growth factors and cytokines should only have the first word capitalized when at the beginning of a sentence.
  8. Intro line 69 “In our previous study,…” insert comma
  9. Intro lines 71 to 73. Should be rewritten “We also found decreasing tear and aqueous MMP9 activity, with increased inflammatory cytokine levels, in severe PXG eyes compared to primary glaucoma[14], suggesting that the elevated TGF levels somehow were important for molecular events that cause the disease…” Note the commas added and the change in other changes to correct English..
  10. Intro lines “75 to 76. “several key miRNA driven by TGF to be upregulated” should be changed to ““several key miRNAs driven by TGF-b to be upregulated…”
  11. Materials and Methods line 89. “phosphate buffer saline (PBS)” there are several PBS formulas reported in the literature. Please specify the formulation and pH since this is a methods paper.
  12. Materials and Methods line 99. “Cultures that reached 50-70% confluence in flasks were serum starved for 24hrs, followed by treatment with TGF-β1…” Note the comma after 24 hrs.
  13. Materials and methods line 101-102. “…72hrs and 96hrs while one set, that was left untreated, was used as control.” Note commas
  14. Materials and methods line 104, change to “…using a previously described procedure.”
  15. Materials and methods line 109. “…non-canonical pathways), cells were pre-treated…” note comma
  16. Materials and methods line 114-115. “Lysates of TM cells treated with TGF-β1, with or without pre-treatment with inhibitors, were used for different experiments” Note commas.
  17. Materials and methods line 120. EMT is most commonly defined as “Epithelial mesenchymal transition” not “Epithelial mesenchymal transformation.”
  18. Materials and methods line 123 to 126. Change tense and add commas. “EMT was indicated when HTM cells became elongated, spindle-like and densely packed losing typical characteristic morphology of HTM cells. These changes were captured using an inverted microscope (Olympus CKX53, Japan), which was confirmed with upregulation of smooth muscle actin, α-SMA and vimentin, which are markers for mesenchymal cells.” Also, α-SMA is more specific for myofibroblast cells. This should be clarified, or it will lead to confusion for readers.
  19. Materials and methods line 129. “we had two criteria's of measurement…” Criteria is pleural. “we had two criteria of measurement. Also, throughout the manuscript the third person should be used rather than using “we” first person pleural. Thus, eliminate “we” throughout the manuscript.
  20. Materials and methods line 170 to 171. Change to “overnight at 4°C, followed by incubation with secondary antibody (anti-rabbit alexa 488 1:2000, Invitrogen, USA) mixed…”
  21. Materials and methods line 175. Change to “The mean data were expressed as…” data is pleural.
  22. Results line 183, begin the sentence with THE. The TGF-β1 signalling pathway is an extremely versatile…
  23. Results line 216. “MMP9 remains an important mechanism of TGF mediated regulation of ECM homeostasis and degradation”. MMP9 is not a mechanism. “modulator” would be better.
  24. Results line 244. “Consistent with our earlier study, we found XBP1 and CCT4…” insert the comma.
  25. Results line 267. “Smad3 inhibition caused reduced TGF induced fibronectin expression, whereas expression of…” insert comma.
  26. Discussion line 292. “without evident glaucom” should be “without evident glaucoma”
  27. Discussion line 303. Change to “…processing prevented visualization of these fibrils.”
  28. Discussion line 306. “…in TM cells at 48hrs, which was also confirmed by increased α-SMA and vimentin expression.” add comma and singular was.
  29. Discussion line 397. Better if changed to, “This paralleled increased IL-6 and IL-8 levels at the same time period, both of which are TGF-induced deleterious effects for ECM homeostatsis.”
  30. Discussion line 309. “…Smad3 inhibition, while other proteins were differentially regulated by other pathways differently. Differentially and differently are redundant. Add comma.
  31. Discussion line 313. Better as “EMT-associated disease pathogenesis has already been studied in a range of diseases including lung fibrosis and cancer.”
  32. Discussion line 314. “In the eye, EMT has been associated with subretinal fibrosis leading to age-related macular degeneration (AMD).” Add comma. “shown to cause” is overstated. Many processes lead to AMD.
  33. Discussion line 316. “The differential effect of recruitment of canonical and non-canonical pathways to regulated different ECM proteins maybe one of the mechanisms of the TGF paradox driving increased fibrosis in advanced disease in eyes with glaucoma.” Very awkward sentence. Reword and correct grammar.
  34. Discussion line 318. “Study of these effects in our model particular could give more insights into actual regulation and cross talk of other pathways regulating TGF induced fibrosis in HTM cells…” Don’t use regulation and regulating in the same sentence. Perhaps modulating for regulating?
  35. Discussion line 320. “It may also identify” better “It may also be used to identify…”
  36. Discussion line 332. Better as “misfolded proteins forming insoluble aggregates.”

Reviewer 2 Report

The authors present an interesting  in-vitro model of pseudoexfoliation syndrome (PXF),  based on primary human trabecular cells   from healthy donors, to analyse several pathways which seem to trigger a fibrillopathy into trabecular environment promoting the onset of glaucoma.

The obtained primary HTM cells were treated with prolonged exposure  to various concentrations of TGF-β1 to induce a fibrotic response. The study design considered several relevant end points:   Cell viability and death index, epithelial mesenchymal transition (EMT)  and related EMT markers, as well pro-fibrotic markers and  Protein complex and amyloid aggregate formation.

The exposure to inhibitors of canonical and non-canonical TGF pathway evidenced that   very complex  pathways are implicated in the reported   different regulation of   EMT and pro-fibrotic expression.

The aims of this paper are relevant to deep the pathogenesis of glaucoma.

To better investigate the real conditions that underly the glaucoma,  including PXF, I suggest   to verify their interesting findings in a dynamic 3D-model  of HTM  to better mimic the real environment
